# The Effect of the Sterile Insect Technique on Vibrational Communication: The Case of *Bagrada hilaris* (Hemiptera: Pentatomidae)

**DOI:** 10.3390/insects14040353

**Published:** 2023-04-02

**Authors:** Chiara Peccerillo, Chiara Elvira Mainardi, Rachele Nieri, Jalal Melhem Fouani, Alessia Cemmi, Massimo Cristofaro, Gianfranco Anfora, Valerio Mazzoni

**Affiliations:** 1Center of Agriculture, Food and Environment (C3A), University of Trento, 38010 San Michele all’Adige, Italy; chiara.peccerillo@unitn.it (C.P.);; 2Biotechnology and Biological Control Agency (BBCA) Onlus, Via Angelo Signorelli 105, 00123 Rome, Italy; 3Department of Environmental Biology, University of Rome “La Sapienza”, 00185 Rome, Italy; 4Department of Mathematics, University of Trento, 38123 Trento, Italy; 5Italian National Agency for New Technologies, Energy and Sustainable Economic Development (ENEA), Via Anguillarese 301, 00123 Rome, Italy; 6Research and Innovation Center, Fondazione Edmund Mach, 38010 San Michele all’Adige, Italy

**Keywords:** sterile insect technique, biotremology, biological control, sexual selection, pest control, product quality assurance

## Abstract

**Simple Summary:**

The sterile insect technique (SIT) is an eco-friendly control strategy, but to implement it effectively, the insects that are subject to sterilization through irradiation should have fitness levels that are comparable to wildtype individuals. This study aims to evaluate the potential use of this technique for the control of *Bagrada hilaris* by investigating the efficacy of courtship in pairs of insects with untreated females and males irradiated at 60 and 100 Gy under laboratory conditions, with a focus on vibrational communication. Males irradiated at 60 Gy and the controls (unirradiated males) exhibit reproductive behaviors and mating abilities that are similar to each other, while insects irradiated at 100 Gy show a severe decrease in the quality of sexual performance.

**Abstract:**

The painted bug, *Bagrada hilaris*, is an agricultural pest in its original areas (Africa, South Asia, and the Middle East), and it has recently been recorded as an invasive species in southwestern part of the US, Chile, Mexico, and two islands in the Mediterranean basin. Its polyphagous diet causes severe damage to economically important crops. The control of this pest is primarily achieved by means of synthetic pesticides, which are often expensive, ineffective, and harmful to the ecosystem. Recent physiological bioassays to assess its potential control through the sterile insect technique demonstrated that mating between untreated females and males irradiated at doses of 64 and 100 Gy, respectively, resulted in 90% and 100% sterility of the eggs produced by the females. In this study, the mating abilities of virgin males irradiated at 60 and 100 Gy with virgin females were measured through a study of short-range courtship mediated by vibrational communication. The results indicate that males irradiated at 100 Gy emit signals with lower peak frequencies, mate significantly less than unirradiated males do, and do not surpass the early stages of courtship. Conversely, males irradiated at 60 Gy present vibrational signal frequencies that are comparable to those of the control and successfully mated males. Our findings suggest that *B. hilaris* individuals irradiated at 60 Gy are good candidates for the control of this species, given that they retain sexual competitiveness regardless of their sterility, through an area-wide program that incorporates the sterile insect technique.

## 1. Introduction

As a consequence of globalization and increasing human population, biological invasions represent one of the most significant challenges in the 21st century [1,2]. Invasive alien species are a threat to ecosystems, affecting biodiversity and human activities.

The case of the painted bug, *Bagrada hilaris* (Burmeister) (Hemiptera: Pentatomidae), a stink bug native to South Asia, Middle East, and Africa [3] is particularly relevant because of the damage that this pest causes economically important crops [4]. Its broad range diet has allowed *B. hilaris* to become a serious concern, in particular for Brassicaceae, but also for many other important crops in the USA [5,6], México [7], and Chile [8]. In Europe, *B. hilaris* is present in the islands of Pantelleria (Italy) and Malta. In Pantelleria, it was first recorded in 1963 on caper (*Capparis spinosa*), and in a short time, it became a key pest of this crop, which is a typical product of the island recognized by a protected geographical designation [9].

*Bagrada hilaris* exhibits a “lacerate-and-flush” method of feeding that causes circular chlorotic lesions to the leaves that eventually result in the destruction of apical meristems, thus blocking growth of the terminal part of the shoot [10,11]. The mechanical feeding damage is amplified by a characteristic gregarious behavior of nymphal instars. In case of severe injuries, the affected host plants could be deemed to be non-commerciable [12]. In all areas where the painted bug is invasive, including the island of Pantelleria, where much of the territory is protected as a national park, the only available control methods involve multiple applications of broad-spectrum synthetic insecticides such as pyrethroids and neonicotinoids, which are effective at containing the population outbreaks and prevent economic damage [13]. However, this chemical control approach is not sustainable due to the development of resistance by the target insect [14], as well as the significant economic costs and ecological impact [15]. Consequently, there is an urgent demand to develop new damage mitigation and prevention strategies that are species specific with no or low off-target impact [16]. To prevent the negative effects induced by these control methods, the sterile insect technique (SIT) included in an area-wide IPM program could be a potential alternative [17]. 

The SIT method is based on the mass breeding or collection of males of the target pest, their sterilization through irradiation, and their release into the environment at regular intervals. Wild females that mate with sterile males do not produce offspring, thus preventing population growth [18]. Reproductive infertility is typically induced by exposure to X-rays, electron beams, or γ-rays generated from Cobalt or Cesium sources [19]. 

Up to now, SIT field applications have been restricted to holometabolous insects, such as Diptera and Lepidoptera [17], because the release of sterile phytophagous heterometabolous adults can lead to unwanted damage to host crop species [20]. Nevertheless, an appropriate application of an SIT program could potentially be effective for suppression and eradication in particular geographic and/or infrastructural conditions (e.g., greenhouses, siloes, and isolated crop areas) if the following conditions related to the target pest biology are met: multivoltinism, gregarious behavior, and the mating behavior is unaffected by irradiation [20]. Therefore, considering the biology of the pest in question and the geographical characteristics of the island, Pantelleria is a suitable area in which SIT could be effective for the eradication of *B. hilaris.*

To improve the chances of eradication, SIT can be used in combination with complementary control techniques [21]. In particular, classical biological control appears to be a compatible strategy to combine with SIT in an AW-IPM program [16]. The release of potential sterile biocontrol agents [22] or the use of sterile eggs as sentinel eggs [23] are some approaches that may work in synergy with it. Classical biological control studies have revealed the existence of some wasps of the Scelionidae (genera *Trissolcus*, *Gryon,* and *Ooencyrtus*) able to intercept and parasitize *B. hilaris* eggs [24,25,26].

Furthermore, in regard to mass breeding processes for SIT, there are no data in the literature describing that of *B. hilaris*. However, it is known that the mass-rearing process can promote genetic drift, inducing genotypic differences between wild and laboratory populations [20,27]. For this reason, a possible alternative is to conduct massive collections of Bagrada bugs to create “small-scale mass-rearing natural bio factories” to provide and release sterile wild-type insects [21,28].

In addition, to evaluate the feasibility of the SIT, it is necessary to carefully monitor the phenotype of reared sterilized males and observe the physical, physiological, and behavioral differences of irradiated males compared to those of the non-irradiated ones in order to assess the optimal radiation dose that ensures a good trade-off between sterility, longevity, and competitiveness.

Recent studies have confirmed that mating females with males irradiated at 64 Gy and 100 Gy results in 90% and 100% sterility of the eggs laid, respectively [23].

Therefore, the successful application of the SIT requires that the reared sterilized males can successfully compete and mate with their wild counterparts [20] and be competent in their ability to communicate with females, as receivers and/or senders of signals. 

Thus, regardless of the total sterility that is reached starting with an irradiation dose of 100 Gy, lower irradiation doses must be considered, as the right balance between sterility and the ability to compete sexually needs to be achieved [29]. In-depth understanding of the sexual communication processes of the pest species is therefore essential to verify this [30].

Vibration-sensitive insects have implemented a language composed of vibrational signals that can serve both to detect the presence of natural enemies or prey and to communicate with individuals of the same species [31]. In the case of *B. hilaris*, it has been confirmed that short-range courtship is mediated not only by cuticular hydrocarbons [32], but also by intra-specific vibrational communication. Both reproductively mature males and females emit signals for partner localization and recognition through wing flapping. Then, the male produces pulse trains with a harmonic frequency structure during the courtship phase, involving the mounting of the female. As a component of the mating behavior, a similar type of signal also occurs during copulation through the male’s abdominal contractions [33].

The present work aims to employ applied biotremology to add basic information on the quality assessment of irradiated *B. hilaris* male subjects to be released into the environment for the implementation of SIT. Consequently, the ultimate goal is to investigate the differences between irradiated and unirradiated males at the pre- and during copula vibrational communication level and contextualize the observational study of the courtship system according to substrate-borne signals.

## 2. Materials and Methods

### 2.1. Insects Rearing and Irradiation

Nymphs and adults of *B. hilaris* were collected in September 2022 from capers (*Capparis spinosa* L.) fields that are most affected by pest population near the village of Scauri (36.7722; 11.0606; 22 m a.s.l.) on the southwestern coast of Pantelleria island. Then, insects were transferred to the quarantine facility at the Edmund Mach Foundation (FEM) in San Michele all’Adige (Italy) and reared in mesh insect cages (30 × 30 × 30 cm, BugDorm^®^, Taichung, Taiwan), in which Brussels sprouts (*Brassica oleracea* L. var. *gemmifera*) were offered as food. The rearing temperature was set at 25 ± 1 °C, relative humidity was 60%, and the light: dark cycle was 16:8. To allow oviposition, one Petri dish (9 cm Ø) filled with fine sand was placed in each cage [34].

Fifth-stage nymphs were isolated individually in plastic 50 mL Corning^®^ Falcon tubes covered with organdie mesh for transpiration. Consequently, after a maximum of 24 h following the final molt, only the males were recovered, and then they were irradiated.

The insects underwent the irradiation process at 60 or 100 Gy (dose rate 2.92 Gy/min) in groups of 10 individuals, which took place at the 60 Co gamma facility at ENEA (Italian National Agency for New Technologies, Energy, and Sustainable Economic Development, Rome) [35].

### 2.2. Experimental Set-up and Signal Recording

The experiments were carried out from October to December 2022 in a soundproof room in the biotremology laboratory of Fondazione Edmund Mach. All replicated experiments were conducted at 25 ± 1 °C, RH = 30–60%, from 9:00 to 18:00.

We recorded the insects in couples of unirradiated females and males or unirradiated females and irradiated males. Each couple was placed on the upper apical leaf lamina of caper plant grown in pot and placed on an anti-vibrational table. In total, 89 couples were tested: the males in 27 couples were irradiated at 60 Gy (T60), the males in 31 couples were irradiated at 100 Gy (T100), and 31 couples included unirradiated males (control). Each trial lasted 15 min and began when both individuals were on the plant. Substrate-borne vibrations emitted by insects were recorded by focusing the beam of a laser vibrometer (VibroGo, Polytec, The Netherlands) on the stem of the plant as close to the leaf as possible. A reflective tape attached to the stem was used to optimize laser reflection. The laser was connected to a computer through a LAN-XI data acquisition system (type 3050-B-040, Brüel & Kjær sound and vibration A/S), and the signal was acquired using “B&K connect” software (Brüel & Kjær sound and vibration A/S). Subsequently, audio files were filtered and analyzed using the software programs Raven Pro 1.6 (The Cornell Lab of Ornithology, Ithaca, NY, USA) and Adobe Audition 23 (Adobe Systems, Inc., San Jose, CA, USA).

### 2.3. Data Analysis

#### 2.3.1. Terminology of Vibrational and Behavioral Parameters

The parameters taken into account to evaluate the effects of radiation on the sexual communication of *B. hilaris* can be organized into two sections: the first concerns the vibrational signals emitted by the male and their distribution in the recordings; the second is purely observational and involves the behaviors showed by the individuals during recordings.

Single signals are isolated vibrational signals, while trains are signals emitted in chain with a repetition time between 0.01 and 0.05 min. The variables measured in the spectral analysis of signals are peak frequency and delta time. Peak frequency is defined as the frequency of the highest peak in a signal’s frequency spectrum, excluding low-frequency noise below 40 Hz [36], and delta time is the duration of the signal in seconds.

In the behavioral study, the mount phase [32] consists of the courtship step, in which the male climbs on the female’s back. If the courtship is successful, the mount phase is followed by mating. In replicates in which no contact between the insects occurred, it is referred to as still behavior. The duration of mounting phase indicates the time interval from the beginning of mounting phase to the beginning of mating. During mating, the male produces abdominal muscle contractions. Their function has not yet been studied, but they probably act to facilitate sperm transfer. Single signals and trains can occur either during mount phase or during mating and are referred to as mounting signals and mating signals.

The time of day was classified into morning, from 10:00 to 14:00, and afternoon, from 15:00 to 18:00. For age, individuals with a number of days less than or equal to 7 were considered to be young; individuals from 8 to 17 days were considered to be old. For simplicity, males irradiated at 100 and 60 Gy are indicated in the results as T100 and T60, respectively.

#### 2.3.2. Statistical Analysis

All statistical analyses were performed on R 4.1.1 (R Core Team 2021, Vienna, Austria) [37]. The data were always checked for normality using a Shapiro–Wilk test before conducting any statistical test. When normality was met, a one-way ANOVA was used to check differences between groups. Non-normal data were analyzed using Wilks’ Lambda type non-parametric interference using the “npmv” package [38] or a Kruskal–Wallis test followed by a Dunn test as a post hoc using the “dunn.test” package [39].

To describe the influence of the treatments on the mating behavior of the insects (i.e., Still, Mount, or Engagement), two generalized linear mixed models (GLMM) with a binomial distribution were fitted using the “glmmTMB” package [40]. Within the first GLMM, we considered the dependent variable “behavior 1”, consisting of “No Mating” and “Mating”. The “No Mating” level comprised the “Still” and “Mount” behaviors. Within the second GLMM, only the “Mount” and “Mating” behaviors were considered to evaluate the effect of the signals produced within the mount phase on the mating success, herein after referred to as “behavior 2”. For both models, the “day period” was used as a random factor. The explanatory factors of the first model were “Treatment” and “Age”, while for the second model, they were “Treatment”, “Age”, and “Mount signals”. Akaike Criterion Information (AIC) [41] was used to choose the best fitting model (Appendix A) after verifying the lack of patterns within the residual diagnostic plots using the package “DHARMa”. All plots were generated using “tidyverse” package [42].

## 3. Results

Both vibrational signals and the mating behavior were affected by the irradiation. In particular, the T100 treatment showed to be the most impactful treatment.

### 3.1. Vibrational Communication

The structure of vibrational signals, both during the mounting phase and mating, were unaltered by the treatments. All signals had a harmonic structure, with a frequency modulation that resulted in an increased dominant frequency at the end of the signal for signals longer than 0.41 s or in hill-shaped spectrogram for shorter signals. 

In T100 males, both the averages of the signals produced in the mounting and mating phases differ from those emitted in T60 and in the control (peak frequency mount signals: chi-squared = 15.78, df = 2, *p* < 0.001; peak frequency mating signals: chi-squared = 52.63, df = 2, *p* < 0.001) by a lower peak frequency (Figure 1); in the mount phase, this was lower by about a dozen Hz, while in the mating, it was lower by more than 20 Hz.

As for the duration of the signals (Figure 2), no significant difference (delta time mount signals: chi-squared = 0.92, df = 2, *p* = 0.63; delta time mating signals: chi-squared = 4.05, df = 2, *p* = 0.13) is reported between the treatments in either the mounting signals or the mating signals.

The median of the averaged numbers of signals produced by the bugs in the mounting and mating phase is presented in Figure 3. Although no statistically significant differences were found between the treatments (chi-squared= 0.63, df = 2, *p* = 0.73), a higher value in the control and the presence of two outliers in T60 and the control is still observable.

### 3.2. Pre-Copulatory and Mating Behavior

The treatment effects discernible with the mount phase are expressed in Figure 4. Bar plot 4a shows that T60 males emit significantly more signals during the mounting phase compared to those of the T100 and control males (Wilks Lambda = 6.340, df = 2, 86, *p* = 0.003). Despite this difference, the number of copulations following the mounting phase were similar in T60 and control males, whereas it was significantly lower in T100 males (Wilks Lambda = 10.394, df = 2, 71, *p* < 0.001). In fact, about 75% of mounting events observed in the T100 did not lead to mating.

Despite the number of replicates in which mating occurred being distinctly different (chi-squared = 13.498, df = 2, *p*-value = 0.001172) among the control (23 of 31), T60 (19 of 27), and T100 (10 of 31), the number of signals emitted during mating was the same among all the treatments (Wilks Lambda = 0.778, df = 2, 49, *p* = 0.465).

The rates of abdominal muscle contractions per minute were highest in the T60 males and lowest in the T100 males (F (2, 23) =7.841, *p* = 0.00253) (Figure 5).

The time interval between the mounting phase and mating (Figure 6) was significantly longer in the T100 males than it was in the T60 and control couples (chi-squared = 6.5299, df = 2, *p*-value < 0.05).

There were no problems detected in the residual diagnostics of the models. The significant factor of behavior 1′s GLMM was treatment 100 Gy (z = −3.222, *p* = 0.00127). The significant factors of behavior 2′s GLMM were treatment 100 Gy (z = −2.7747, *p* = 0.000888) and a young age (z = 1.642, *p* = 0.017961) (Table 1). 

The coefficients and *p*-values returned by the first GLMM (Figure 7a) indicate that T100 males had a significantly lower probability of mating than the untreated males did. The results obtained from the second GLMM (Figure 7b) did not detect a significant effect of having a signaling activity during the mounting phase on the occurrence of mating. Unlike the first GLMM, in addition to T100, age also appeared to have a significant impact, suggesting that younger insects were more likely to mate following the mount phase than the old ones were.

## 4. Discussion

The product quality assurance is a key process, in which the insects to be released are evaluated for effectiveness in completing the purpose for which they are treated [43]. Evaluating the quality of an insect for the SIT involves knowing every aspect of radiation that may have some impact on their performance and competitiveness with wild insects.

To date, SIT has seen its successful application in insect groups, such as Diptera and Lepidoptera, which use vision [44] and chemical signals [45] as the main components of intraspecific communication. Therefore, the parameters considered for the study of the effects of irradiation on communication have so far only taken into account the components mentioned above [46].

In the case of *B. hilaris,* it is an insect that does not solely rely on semiochemicals for sexual communication purposes; therefore, the product quality assurance needs to be fine-tuned by adding other parameters. In our own case it can be said: with new insects, there need to be new criteria. Indeed, this study highlights the importance of considering the ability to emit and receive vibrational signals in pentatomids to identify any communication and courtship deviations from those of the wildtype phenotype.

Overall, the results obtained in this work support the hypothesis that the most pronounced differences with control individuals are represented by males that were irradiated at 100 Gy. Previous observational studies have investigated the reproductive behavior of *B. hilaris*, showing that males irradiated at 60 Gy show no significant decrease in mating ability [47].

This result is confirmed in the present work, as in our experiment, the mating success of males irradiated at 60 Gy is comparable to that of the controls. However, the males treated at 60 Gy show both a significant increase in vibrational signals during the mounting phase and a significantly higher frequency of abdominal muscle contraction during engagement with respect to those of the untreated controls. These intriguing results suggest a better reproductive performance in treated males compared to that of the controls. It is still unclear how radiation may positively influence some physiological (sperm number, mortality, and/or viability) [48,49] and behavioral traits (frequency and duration of “pheromone-call”) (C. O. Calkins and T. R. Ashley, unpublished data), as long-term monitoring of the irradiated phenotype is needed to better understand this phenomenon.

The overall analysis of the short-range communication and behavior of the pairs with males treated at 100 Gy shows that the sexes are able to recognize and locate each other and make contact. The subsequent rejection of the male by the female, and thus, the failure to mate can therefore also be attributed to a modification of the vibrational signals emitted by the male following treatment at this dosage. In particular, the differences with untreated males do not seem to involve the quantity of signals emitted by treated males, but rather their quality and structure. Indeed, compared with untreated individuals, a lower average peak frequency during mounting has been revealed, which could be a reason for rejection by the female. We can therefore assume that despite being able to induce total sterility [28], irradiation at 100 Gy is likely to reduce the competitiveness of the insect and preclude a sufficient probability of mating with wild females; thus this method is unsuitable for use in an SIT program.

It is essential to keep in mind that vibrational signals are only one of many ways in which *B. hilaris* communicates with individuals of the same species: interspecific communication system of the painted bug is not limited to the use of vibrations, but it also involves the use of volatile and contact pheromones to promote gregarious behavior in nymphs and reproduction in adults [32,50]. In this regard, it is necessary to refer to applied biotremology as a useful tool for assessing the quality of individuals only if it is complemented by other considerations. The effects of gamma rays can also potentially be detected at the level of chemical communication, and future studies are needed to provide a complete overview of radiation-induced physiological and behavioral changes. Finally, our results are encouraging to apply the SIT method to other problematic pentatomids, such as the brown marmorated stink bug *Halyomorpha halys* (Stål). This pest has already been subject to physiological and behavioral studies on the effect of irradiation on its fertility and competitiveness [51], and its vibrational communication was described previously by Polajnar et al. (2016) [36]. The investigation of potential radiation-induced differences in vibrational communication could therefore bring refinement to the quality evaluation of this species too. 

## 5. Conclusions

The results obtained highlight that irradiation doses at 100 and 60 Gy have different impacts on the fitness of individuals. In the former case, the vibrational signals emitted have lower frequencies, and mating success is drastically impaired, while males treated at the lower dose maintain a sexual performance that is comparable to that of unirradiated individuals. These findings lead to the conclusion that *B. hilaris* individuals irradiated at 60 Gy are good candidates for the control of this species, given that they retain sexual competitiveness regardless of their sterility, through the use of SIT in an AW-IPM program.

## Figures and Tables

**Figure 1 insects-14-00353-f001:**
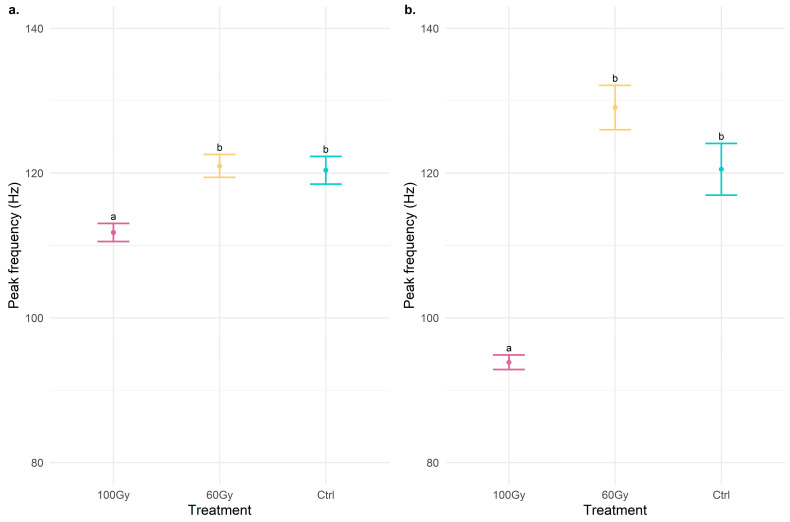
Dot plots with error bars of the peak frequency (Hz) of signals emitted during (**a**) the mount phase and (**b**) the mating. The dot represents the mean of peak frequency, and the error bar represents the standard error of the mean. The treatments indicated with different letters show significant differences (Kruskal–Wallis test at *p* < 0.001).

**Figure 2 insects-14-00353-f002:**
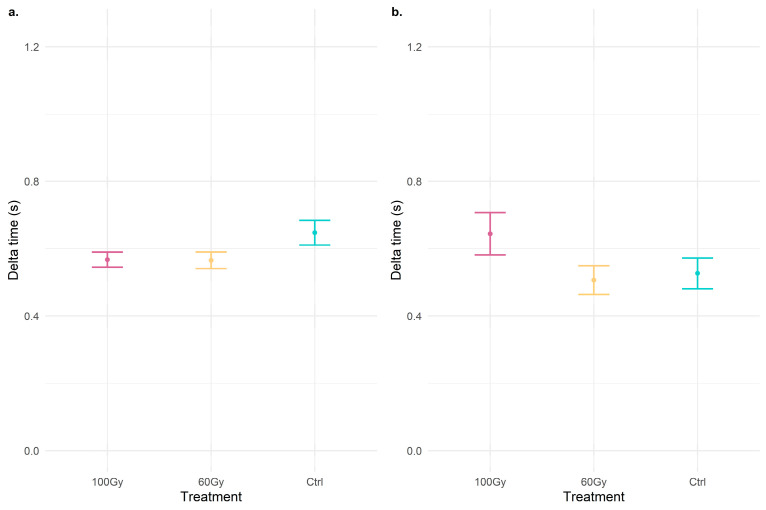
Dot plots with error bars of the delta time (s) of signals emitted during (**a**) the mounting phase and (**b**) mating. The dot represents the mean of delta time, and the error bar represents the standard error of the mean.

**Figure 3 insects-14-00353-f003:**
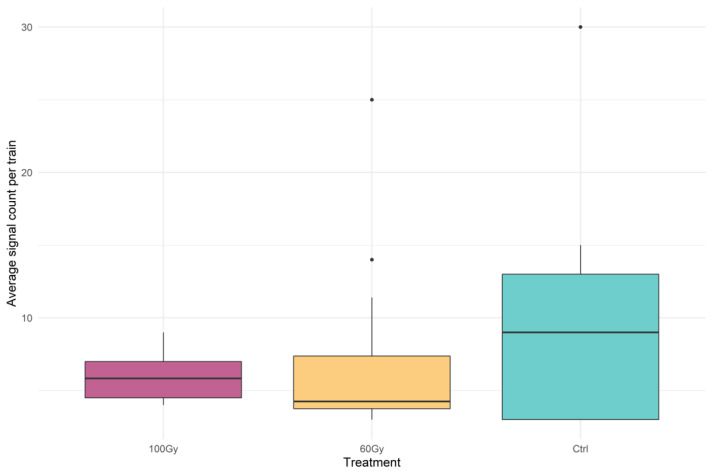
Box plots of the average signal count per train for each treatment. The box plots represent the interquartile range, with a horizontal bar being the median. The bottom whisker and the upper whisker represent the minimum and maximum values, respectively. The dots show the outliers.

**Figure 4 insects-14-00353-f004:**
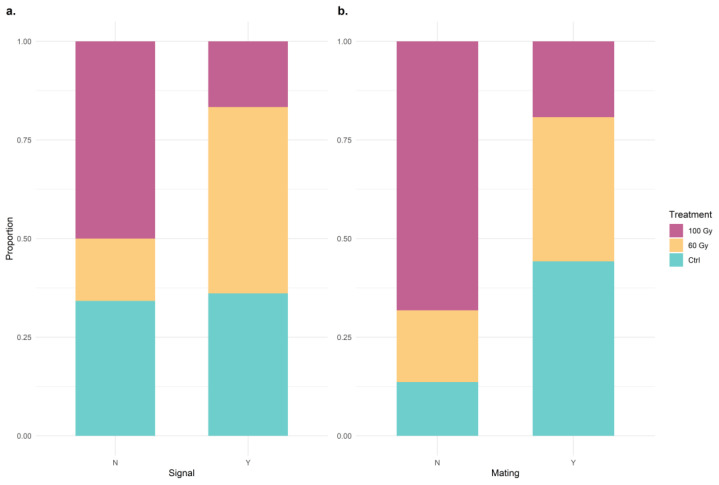
One hundred percent stacked column chart of the effect of different radiation grades (**a**) on the occurrence of signals during the mount phase and (**b**) on mating success following mount phase for each treatment. “N” stands for No, and “Y” stands for Yes. Only the individuals that have reached the mount phase were considered for this analysis.

**Figure 5 insects-14-00353-f005:**
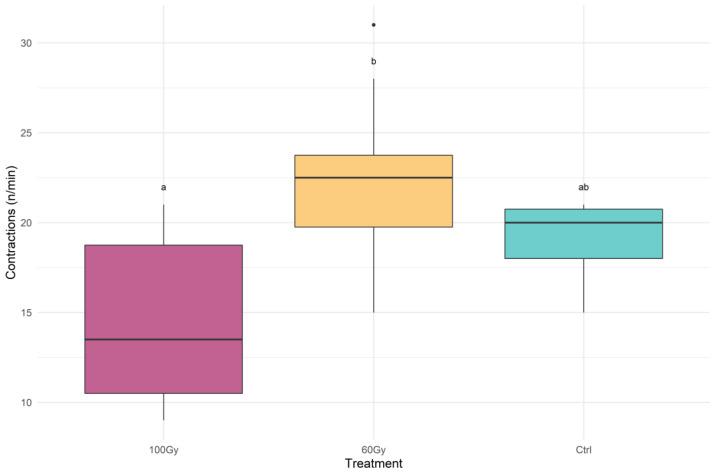
Box plot with the rate of contractions per minute that occurred during mating for each treatment. The treatments indicated with different letters show significant differences (one-way ANOVA at *p* < 0.01). The box plots represent the interquartile range, with a horizontal bar as the median. The bottom whisker and the upper whisker represent the minimum and maximum values, respectively. The dots show the outliers.

**Figure 6 insects-14-00353-f006:**
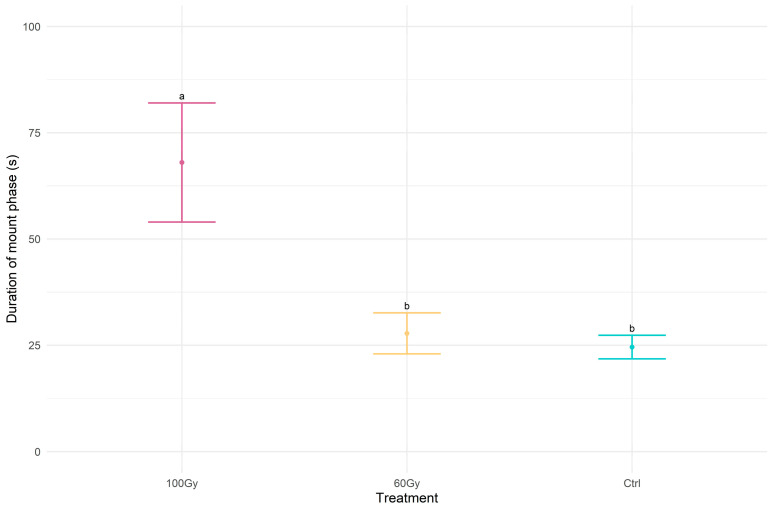
Dot plot with error bars of the time interval counting from the beginning of the mount phase to the beginning of mating. The treatments indicated with different letters show significant differences (Kruskal–Wallis test at *p* < 0.05). The dot represents the mean of delta time, and the error bar represents the standard error of the mean.

**Figure 7 insects-14-00353-f007:**
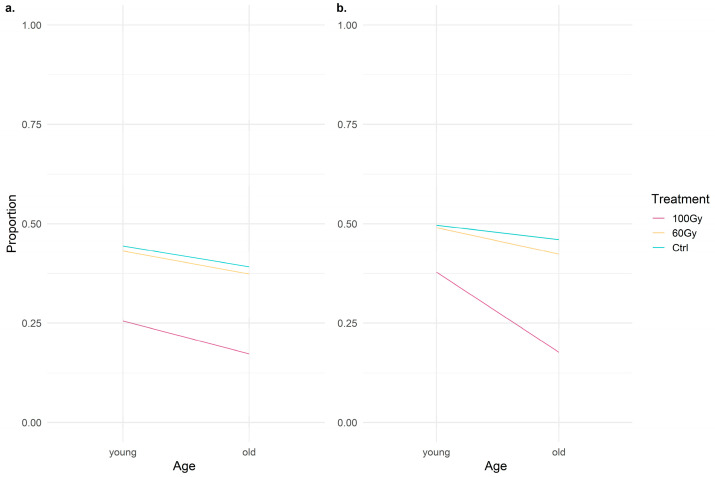
Line plots describing at all three doses the probability of the occurrence of (**a**) behavior 1 (i.e., no mating or mating) and (**b**) behavior 2 (i.e., Mount or Mating) between young and old insects.

**Table 1 insects-14-00353-t001:** Results of generalized linear mixed-effects model (GLMM) for mating and mount signals (m.sig). (A) The best explanatory model for each dependent variable. (B) The significant independent variables of the chosen models. Dev (deviance), Est (estimate), df (df residuals). * *p* < 0.05, ** *p* < 0.01, *** *p* < 0.001.

A	Model Synthesis	AIC	BIC	LogLik	Dev.	df
(mating)	behavior1 ~ treatment + age+ (1|day period)	115.3	127.7	−52.6	105.3	84
(m.sig)	behavior2 ~ treatment + age+ Mount signals + (1|day period)	79.0	92.8	−33.5	67.0	68
**B**	**Fixed Effects**	**Estimate**	**Std. Error**	**Z Value**	**Pr (>|z|)**	
(mating)	(Intercept) treatment 100 Gy	0.7850 −1.8429	0.4512 0.5721	1.740 −3.222	0.08188 0.00127 **	
(m.sig)	(Intercept) Treatment 100 Gy Age Young	1.7371 −2.7747 1.6420	0.7383 0.8348 2.366	2.353 −3.324 2.366	0.018627 * 0.000888 *** 0.017961 *	

## Data Availability

The data presented in this study are available on request from the corresponding author.

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
