# Peer review of "The Effect of the Sterile Insect Technique on Vibrational Communication: The Case of Bagrada hilaris (Hemiptera: Pentatomidae)"

_insects, 2023, doi:10.3390/insects14040353_

Round 1

Reviewer 1 Report

This manuscript describes a series of well designed and appropriately analysed experiments which address questions in an emerging area of research. That is, the use of SIT in insects that use vibrational communication, and the impact that irradiation has on mating behaviour and therefore the efficacy of an SIT programme. Whilst the results are likely only useful to a small group of researchers, the results are very important, and interesting beyond the scope of B.hilaris. The context for the study, methods, results, conclusions, and relevance are all clearly described. My only big issue with the paper was that the figure legends were far too brief. Other than that, I have only a few minor comments to offer below:

L28 – allows instead of allowed.

L51 – Its instead of The

L75 – s on agent

L78 – sentence needs re-wording.

L84 – remove the

L95 – this sentence doesn’t quite make sense. Maybe put a “it is deemed” after considering?

L100 – reword to “To evaluate the feasibility of the SIT…. “

L111 – replace how with that? Something wrong with this sentence

L124 – italicise two scientific names in this sentence (and later in methods brussel sprouts)

L146 – missing a d on irradiate.

L150 – do full stop, not semicolon.

L182 – Shapiro-Wilk capitalise.

L214 – You need to specify what your error bars are representing e.g. standard error of the mean or other. This goes for all of your graphs.

L214 – Your figure legends need to be much more descriptive. They should be able to stand alone from the text of the paper. This goes for all of your graphs.

L299 – What are you saying here – it is not clear.

Author Response

We are grateful for the time and effort you took to review our work and provide such thoughtful and insightful feedback. We thank you for the constructive corrections and suggestions that helped us to improve the overall quality of the manuscript.
Please see the attachment for specific comments.

Reviewer 2 Report

The study is scientifically sound and well presented. Both the target species and the SIT method are important for pest management.

I suggest the authors to explain in the introduction if and how, is possible to rear B. hilaris in large numbers to be feasible to perform SIT for this species. If this is not studied yet, then highlight it as an important element for the SIT.

Since authors refer to the biological control I also suggest to explain a bit more the compatibility of SIT with biological control (see Kapranas et al 2022, Entomol Exp. Applicata.)

Check for typos in the text.

Line 65, add “as” such as

Lines 67-69. Does the reference 13 refer to reported resistance of B halaris to insecticides? If not, remove the sentence.

Line 143, please define healthy,

Line 170, replicates

Figure 4, explain what is N and Y

Author Response

(The authors gave the same response as above.)

Reviewer 3 Report

The paper explores the intriguing possibility of using the SIT technique to control an invasive pentatomid species. The experiments are well-conducted, and the data is analyzed fairly. However, several flaws are present regarding mainly the scientific approach.

The main concerns and weaknesses evidenced are listed below:

·        In the introduction and discussion could benefit from greater emphasis on contextualizing the technique and highlighting the importance of vibrational signals for this species.

·        In SIT's previous work, it was shown that sterility was induced starting from an irradiation with 100 Gy. Now, authors are trying to see if these males can compete with wild ones, but with a too narrow approach, as if in nature there were no other signals that mediate mating, and as if Bagrada did not live in dense populations, which is why there will likely be many signals coming from more than one male and female at the same time.

·        In the conclusions, however, they revisit the effect of irradiation, reporting that individuals irradiated with 60 Gy are good candidates for SIT, when in the previous work mentioned here in a very evasive manner, it was stated that the candidates for induced sterility are those at 100.

·        Another main concern is that authors overlooked that it's important to remember that other stimuli rather than vibrational are likely to play a role in the sexual and aggregation behavior of this species, as well as in other pentatomid species.

·        The lack of strong results brings the authors to in the end change the paper purpose compared to the title, as it becomes about determining whether biotremology can be a means for assessing Product Quality Assurance (lines 322-325).

·        Some parts of the manuscript suffer from an unclear English style, and would benefit from review by a native English speaker.

·        Simple summary is introducing the study but is not giving the main highlights of the work done.

·        Abstract needs to be improved as is too long in background and too short in result

·        In the Materials and Methods section, important details such as the number, modalities, and duration of behavioral observations are lacking.

Some specific comments:

Line 24, change “is” with “was”

Line 25, delete the first sentence of the abstract.

Line 32, …toward male individuals or both sexes? These aspects should be more contextualized.

Line 49-51, revise English style.

Line 60, the impact on the plant determined by the insect has been investigated also by Guarino et al (2017)

Line 63-67, this part should be rewritten. Here is an alternative: “The painted bug invasion has spread to all areas, including Pantelleria Island, where much of the territory is protected as a national park. Unfortunately, the available control methods involve multiple applications of broad-spectrum synthetic insecticides like pyrethroids and neonicotinoids, which are necessary to contain the population outbreaks and prevent economic damage.”

Line 60, revise.

Line 69, delete space.

Lines 73-75, the mention of biological control agents seems out of context here; it should be deleted or better contextualized.

Line 82-83, The authors should place greater emphasis on the fact that SIT typically relies on sterilizing male insects. Furthermore, an essential aspect of SIT, which is often overlooked, is ensuring that the irradiated and sterilized individuals remain competitive for mating compared to healthy individuals found in the field.

Lines 90, there are not many cases of successful SIT on such kind of insects, but only preliminary experiments, so the sentence “…the opportune application of a SIT program is capable of achieving suppression…” should be rewritten and dampened.

Lines 108-109, studies of this topic have been addressed, see literature

Lines 112-115, in consideration that B. hilaris chemical communication has also been described (see literature) and the sexual behavior is then probably mediated by a complex cohort of both chemical and vibrational signals. It would be helpful if the authors could explain why they specifically focused on the irradiation effect on vibrational communication, rather than examining the effect on pheromone production and perception.

Line 124, revise species name in italic here and across the section.

Line 132, add “to allow oviposition”

Line 133-135 virgin adults or (sexed maximum one day after the final molt)?

Line 143, revise: is unclear what was tested. Moreover, were caper plants reared in pot, were they furnished by a nursery?

Line 147. What is the feeding status, starved or not? ...and also what is the host plant rationale? The experiments were done with caper leaf but in the rearing authors use Brassica oleracea L. var. gemmifera. Authors should use the same food in the rearing. Explain why.

Line 149, was a blank experiment carried out?

Line 204, the authors should first mention what was altered.

Line 234-235, this is both and interesting but also a weird result, why some parameters as muscle contraction increase in irradiated males? This should be discussed later carefully in the discussion section.

In figure 1 and 2, what a and b stand for?

In the figure 5 and 6, what are the different letter indicating? and which text was conducted? Please report this in the caption.

Line 292-293, The literature suggests that chemical communication plays a role in the aggregation and sexual behavior of this bug. If the authors hold a different perspective, it would be helpful for them to explain their reasoning. Alternatively, the sentence could be rephrased to reflect a more nuanced perspective.

Line 301, missing dot.

Line 307-309, this sentence is too speculative.

Lines 312-314, I recommend to dampen this sentence.

Author Response

We are grateful for the time and effort you took to review our work and provide such thoughtful and insightful feedback. We thank you for the constructive corrections and suggestions which has helped us to improve the quality manuscript and we are confident that your input will make a significant contribution to the final product. All suggestions (except one, as a matter of style) were accepted.
Please see the attachment for specific comments.

Round 2

Reviewer 3 Report

In general, the paper has seen improvement, as the authors have addressed or provided explanations for the majority of the queries.

Line 63, island not in capital letter

Lines 350-352, This is interesting. However, it may be helpful to provide additional context for the sentence, as it could be interpreted as an attempt to include another (self) citation in the authors' work.

Author Response

In addition to accepting both suggestions provided (L63 and L352), we have fixed the English in some parts. Once again, thank you for your time.